# Structural validity and reliability of the patient experience measure: A new approach to assessing psychosocial experience of upper limb prosthesis users

Linda J. Resnik[1,2]*, Mathew L. Borgia[1], Melissa A. Clark[2,3], Emily Graczyk[4,5], Jacob Segil[6], Pengsheng Ni[7]

1 Research Department, Providence VA Medical Center, Providence, RI, United States of America, 2 Health Services, Policy and Practice, Brown University, Providence, RI, United States of America, 3 University of Massachusetts Medical School, Worcester, Massachusetts, United States of America, 4 Department of Biomedical Engineering, Case Western Reserve University, Cleveland, OH, United States of America, 5 Research Department, Louis Stokes Cleveland VA Medical Center, Cleveland, OH, United States of America, 6 Research Department, Rocky Mountain Regional VA Medical Center, Aurora, CO, United States of America, 7 Boston University, Boston, Massachusetts, United States of America

* Linda.Resnik@va.gov

**Data Availability Statement:** Data cannot be shared publicly because of restrictions imposed by the Department of Veterans Affairs. Data are

## Abstract

Recent advances in upper limb prosthetics include sensory restoration techniques and osseointegration technology that introduce additional risks, higher costs, and longer periods of rehabilitation. To inform regulatory and clinical decision making, validated patient reported outcome measures are required to understand the relative benefits of these interventions. The Patient Experience Measure (PEM) was developed to quantify psychosocial outcomes for research studies on sensory-enabled upper limb prostheses. While the PEM was responsive to changes in prosthesis experience in prior studies, its psychometric properties had not been assessed. Here, the PEM was examined for structural validity and reliability across a large sample of people with upper limb loss (n = 677). The PEM was modified and tested in three phases: initial refinement and cognitive testing, pilot testing, and field testing. Exploratory factor analysis (EFA) was used to discover the underlying factor structure of the PEM items and confirmatory factor analysis (CFA) verified the structure. Rasch partial credit modeling evaluated monotonicity, fit, and magnitude of differential item functioning by age, sex, and prosthesis use for all scales. EFA resulted in a seven-factor solution that was reduced to the following six scales after CFA: social interaction, self-efficacy, embodiment, intuitiveness, wellbeing, and self-consciousness. After removal of two items during Rasch analyses, the overall model fit was acceptable (CFI = 0.973, TLI = 0.979, RMSEA = 0.038). The social interaction, self-efficacy and embodiment scales had strong person reliability (0.81, 0.80 and 0.77), Cronbach's alpha (0.90, 0.80 and 0.71), and intraclass correlation coefficients (0.82, 0.85 and 0.74), respectively. The large sample size and use of contemporary measurement methods enabled identification of unidimensional constructs, differential item functioning by participant characteristics, and the rank ordering of the difficulty of each item in the scales. The PEM enables quantification of critical psychosocial impacts of

available from the Providence VA Medical Center Institutional Data Access/Ethics Committee for researchers who meet the criteria for access to confidential data. Individually identifiable data, excluding Veterans' name and 38 USC $7332-protected information, will be shared pursuant to a written request and IRB approved waiver of HIPAA authorization, with the approval of the Under Secretary for Health, in accordance with VHA Handbook 1605.1 $13.b(1)(b) or 13.b (1)(c) or superseding versions of that Handbook. Please contact Val Micucci@va.gov for more information.

**Funding:** This study was supported in the form of funding by the Department of Veterans Affairs Rehabilitation Research (Grant Nos. VA RR&D A9264A-S and VA RR&D A2936-R) awarded to LR, and (Grant No. RR&D 1 101RX003282-01A1) awarded to JS.

**Competing interests:** The authors have declared that no competing interests exist.

advanced prosthetic technologies and provides a rigorous foundation for future studies of clinical and prosthetic interventions.

## Introduction

New technologies and surgical techniques are being developed to enable higher degree of freedom movement, more intuitive control, and restoration of sensory feedback to upper limb prosthesis users, and these advances have the potential to greatly improve quality of life. However, these recent advancements are also associated with greater risks, higher costs, and longer periods of rehabilitation. Therefore, it is important to understand the benefits and limitations of new prosthetic technologies using validated outcome measures to help clinicians and patients make decisions about prosthesis options.

Measuring the impact of advanced upper limb prosthetic systems is challenging, given that existing measures have not been validated Further, they focus on prosthesis function rather than psychosocial outcomes. Psychosocial factors are known to impact the rehabilitation of persons with amputation. Individuals with upper limb loss have higher rates of depression, anxiety, and body image disorders [1–4]. Body image anxiety has been reported to be more prevalent in persons with upper as compared to lower limb amputation, and more prevalent in women. While several psychosocial measures have previously been developed, such as the Prosthesis Evaluation Questionnaire [5] and the Trinity Amputation and Prosthesis Experience Scales [6], and the Amputee Body Image Scale [7] most focus on lower-limb amputation rather than the unique factors relevant to upper limb loss, and none were developed specifically to assess new and emerging technologies.

Thus, novel, validated instruments to quantify the benefits of these advanced prosthetic interventions are needed to inform regulatory and clinical decision-making.

The Patient Experience Measure (PEM) was first developed for a prior study testing an early prototype prosthesis sensory restoration system (SRS) [8, 9] Initial items related to key constructs were generated and grouped into scales based on the investigative team's assumptions about key constructs measured by the items. The original scales addressed the key constructs of self-efficacy, embodiment, body image, efficiency of the prosthesis, and social touch. The initial PEM was administered during two home studies of an SRS consisting of a sensorized single degree-of-freedom myoelectric prosthesis and implanted neural interfaces for sensory neurostimulation [8, 9]. Study results suggested that the PEM was a responsive measure and was sensitive to the changes induced by SRS.

While these were promising results, the psychometric properties of the PEM had not been assessed and the measure was not refined or validated. The initial PEM scales were created by the investigative team based upon their assessment of each items' conceptual fit with latent constructs of self-efficacy, embodiment, body image, efficiency of the prosthesis, and engagement of the prosthesis in social situations, but fit within these proposed scales had not been quantitatively assessed. Additional research was needed to assess the structural validity and fit of items within the scales. Further, additional data on reliability and normative values in populations of upper limb amputees using traditional, commercially available prostheses was needed for interpreting results of future studies of advanced prosthetic technologies. Thus, the purpose of this study was to refine the PEM and conduct a psychometric analysis of its properties in a population of upper limb amputees using a range of widely available prostheses.

## Methods

### Modifications to the PEM

Modification and testing of the PEM was conducted in three phases: initial refinement and cognitive testing, pilot testing, and field testing (all briefly described below). All phases of the study were approved by the appropriate institutional review boards. All participants received mailed information about the study and gave verbal informed consent. The Institutional Review Board approved use of verbal consent because the study was a nationwide telephone survey; verbal consent (or lack of consent) was documented by the telephone interviewer at the time of telephone contact with the participant.

Prior to cognitive testing, the research team revised the original PEM items and instructions to improve clarity and content. These revised PEM items were administered during cognitive interviews [10, 11] with 20 participants (19 prosthesis users, 1 non-user, 2 bilateral amputees; 50% male; mean age 55.5), as shown in Table 1. There are no definitive rules for determining the appropriate sample size for cognitive interviewing. We conducted cognitive-based interviews until we were unable to identify any new content related to the PEM target constructs, or any major problems with particular questions, response options, or instrument instructions.

During cognitive interviews, participants were asked to think out loud as they answered the items and to identify any instructions or words that were confusing and any questions that were difficult to answer. The items, response categories, and instructions were iteratively refined based on feedback and comments made during these interviews. The participant who was not a prosthesis user only completed the 9 items pertaining to general attitudes and feelings about their body. After cognitive testing was completed, the PEM item content and format were reviewed by experts at our collaborating survey center, and additional refinements were made. Detailed information on the item generation and refinement process is contained in S1 Appendix.

The PEM items were then pilot tested with 18 additional participants (13 unilateral, 4 bilateral, 15 prosthesis users; 50% male; mean age 61.9) (Table 1). The sample size for pilot testing was based on available resources and study timeline. Minor refinements were made after reviewing pilot testing results. The final version of the measure that was used in field testing is shown in S2 Appendix.

Field testing was conducted through a telephone survey. Participants were recruited from an earlier Department of Veterans Affairs (VA) study, a list of persons who had received care at the VA between January 1, 2016 –June 1, 2019, emails sent from the Amputee Coalition of America, and recruitment letters sent from a private prosthetics service company. Prior to data analysis, several items (e.g. shy in public, different from others) were reverse coded so that higher scores across all items indicated better patient experience.

### Factor analysis

Exploratory factor analysis (EFA) was used to discover the underlying factor structure of the PEM items. EFA was conducted using data from the first 320 participants in the study (subsample 1) to facilitate the analytic process. We determined the number of unidimensional factors by assessing the number of eigenvalues >1 and applying parallel analysis. Items were grouped into proposed scales based on factor loadings (>0.3) and conceptual fit with other items within the factors. We examined the unidimensionality of proposed scales by calculating the ratio of the first and second eigenvalues. Ratio values greater than 4 were considered evidence of unidimensionality.

**Table 1. Participants in the cognitive testing and pilot studies.**

| | Cognitive N = 20 | Pilot N = 18 |
|---|---|---|
| | Mn (sd) | Mn (sd) |
| **Age (mn, sd)** | 55.5 (13.4) | 61.9 (15.1) |
| | N (%) | N (%) |
| **Gender** | | |
| Male | 10 (50.0) | 9 (50.0) |
| Female | 10 (50.0) | 9 (50.0) |
| **Amputation level** | | |
| Transradial/wrist disarticulation | 10 (50.0) | 11 (61.1) |
| Transhumeral/elbow disarticulation | 8 (40.0) | 4 (22.2) |
| Shoulder | 2 (10.0) | 3 (16.7) |
| **Bilateral upper limb loss** | 2 (10.0) | 4 (22.2) |
| **Prosthesis User** | 19 (95.0) | 15 (83.3) |
| **Primary* prosthesis type** | | |
| Body-powered | 9 (45.0) | 6 (33.3) |
| Myoelectric | 6 (60.0) | 6 (33.3) |
| Hybrid | 0 (0.0) | 1 (5.6) |
| Cosmetic | 3 (15.0) | 1 (5.6) |
| Sports/recreation | 0 (0.0) | 1 (5.6) |
| Unknown | 2 (10.0) | 3 (16.7) |
| **Etiology (may be more than one)** | | |
| Combat injury | 1 (5.0) | 2 (11.1) |
| Accident | 7 (35.0) | 8 (44.4) |
| Burn | 1 (5.0) | 2 (11.1) |
| Cancer | 2 (10.0) | 1 (5.6) |
| Diabetes | 0 (0.0) | 0 (0.0) |
| Infection | 3 (15.0) | 1 (5.6) |
| Congenital | 4 (20.0) | 5 (27.8) |
| Other | 2 (10.0) | 2 (11.1) |
| **Race** | | |
| White | 17 (85.0) | 12 (66.7) |
| Black | 2 (10.0) | 1 (5.6) |
| Other | 1 (5.0) | 3 (16.7) |
| Unknown | 0 (0.0) | 2 (11.1) |

*Primary type of prosthesis is the prosthesis type used most often. Some participants used more than one type of prosthesis.

A confirmatory factor analysis (CFA) using the proposed scales identified through EFA was conducted to verify the factor structure of the data from subsample 1. We evaluated CFA model fit using the comparative fit index (CFI), Tucker–Lewis Index (TLI), root mean square error approximation (RMSEA), and residual correlations. Values of 0.90 or higher were considered acceptable for CFI and TLI, and values of <0.10 were acceptable for RMSEA. CFA of the proposed scales was then repeated with data from 357 persons (subsample 2). Tucker's Congruence Coefficients (TCC) were used to examine the similarity of the factor structures in subsamples 1 and 2. Factor structures were considered 'equal' (TCC≥0.95), 'good similarity' (0.95>TCC≥0.85) or 'not similar' (TCC<0.84). Next, we fit the CFA, examined unidimensionality (considered acceptable if the $1^{st}$:$2^{nd}$ eigenvalues ratio was >4 or the $2^{nd}$ eigenvalue

was <1), and examined the residual correlation matrix. We identified residual correlations greater than 0.2 and removed items based on the item content [12–14]. MPlus software [15] was used to conduct EFA and CFA.

## Rasch analyses

We used Rasch partial credit modeling (PCM) to evaluate monotonicity, fit statistics, as well as the magnitude of differential item functioning (DIF) by age, sex, and prosthesis use for all items. Rasch analyses involves probabilistic modeling of a latent trait, where persons and items are measured on the same interval scale. We first examined item category response curves to assess monotonicity, i.e. whether item response categories were properly ordered. Response categories that were disordered were collapsed and scales were calibrated with Rasch partial credit models. We identified and dropped items with inlier-pattern-sensitive fit statistic (infit mnsq) values less than 0.6 or greater than 1.4. Items with higher expected infit values suggest that an item may be capturing a different construct; whereas items with low infit values are considered redundant. Residual factor analysis assessed the variance in observations explained by the scale and the unexplained variance in the first contrast of the principal component analysis. When at least 40% of the variance was explained by a scale and an eigenvalue was less than 2 for the 1st contrast, we considered the scale as unidimensional [16]. We re-examined local dependence between items by examining the standardized residual correlation of item pairs with WINSTEPS [17]. We considered values greater than 0.4 as indicating violation of local dependence.

We used two methods to evaluate presence of DIF by age group (>65 vs ≤65), prosthesis use (yes or no), gender (male vs. female), and laterality (unilateral vs. bilateral amputation). The first approach used WINSTEPS to identify DIF based on whether DIF contrasts were greater than 0.64 or greater than $2SE+0.43$ [18]. The second approach, the Lasso method, used the R package GPCMLasso [19]. The Lasso method applies the partial credit models with lasso regularization to penalize the absolute value of regression coefficients of item-specific covariates (e.g. age, sex and prosthesis usage). The optimal tuning parameter of the penalization term was determined based on the lowest Bayesian information criterion value [20]. Items with moderate to severe DIF as identified by the first method and confirmed by the Lasso method were split into separate items for the relevant demographic groups.

Rasch item-person maps were developed for each factor. These evaluated how the range and position of item difficulties corresponded to the range and position of the person score that was generated from all items within each scale. The item-person maps identified the levels of person ability that had a 50% probability of selecting each item's response category (as compared to any higher category) for each item. We then compared these item difficulty levels to a histogram of scores for the sample.

We assessed whether more generalized models (e.g. generalized partial credit model, GPCM) estimated the item response more accurately than PCM. GPCMs relax the assumption of uniform discriminating power and allow for varying slope parameters. We applied 10-fold cross-validation. We iteratively selected one sample as the validation sample and the other 9 samples as training samples, fit the GPCM (or PCM) on the training sample, and applied the estimated item parameters to calculate the expected item responses on the validation sample. We calculated the root mean square difference (RMSD) and mean bias of expected and observed item responses across the validation samples, and compared the values generated from GPCM and PCM. There is no criteria of RMSD (or bias) to determine one model is better than another one [21, 22]. We determined the model results were comparable if the values

of RMSD difference from the different models were less than a half point of the item score (0.34).

Person and item reliability was evaluated using Rasch models. High person reliability ($\geq$0.7) indicates that those individuals with the highest scores truly have the best ability, and high item reliability ($\geq$0.7) indicates that those items rated at the highest difficulty are truly the hardest items.

The Rasch test information function was used to determine the ranges of person scores with reliability $\geq$0.8. This was defined as the test information function being greater than the target information value $\frac{1}{(1-r)V}$, where $r$ is the reliability we want to achieve, and $V$ is the person score variance. The test information function is a measure of the information about a scale's construct that is provided by item responses. For DIF items, we averaged the threshold parameters across groups to create one set of parameters for DIF items. We added a bar plot in the item map to indicate the score range with reliability greater than 0.8. Cronbach's alpha was used to assess internal consistency of items in the final scales (after splitting items with DIF). Following the Rasch and DIF analyses, we used CFA to estimate final fit indices and evaluate correlation between factors.

### Transformation scoring and floor and ceiling

Rasch summary scores are calculated on a logit scale for each of the identified scales. Person logit scores were then standardized into a T-score matrix for the sample, and conversion scoring tables were created to calculate T-scores for future respondents with no missing data.

We examined score distributions to evaluate the extent of floor and ceiling effects in our sample. We considered floor and ceiling effects to exist in a scale if 15% or more of the sample had the lowest or highest (item or T-score) scores possible.

### Test-retest reliability and minimal detectable change

Shrout and Fleiss intraclass correlation coefficient (ICC) type 3,1 was calculated using data from the 50 participants who completed the survey twice within 2 weeks to assess test-retest reliability [23]. The minimal detectable change (MDC) at 90% and 95% confidence was estimated using ICC and pooled standard deviation of factor scores (at both time points).

### Results

Characteristics of participants in the field test subsamples 1 and 2, and the subset who participated in the test-retest reliability sample, are shown in Table 2. The full sample age was 61.3 (sd 14.6) years old, on average, with amputations that had occurred a mean of 29.1 years (sd 19.6) prior. On average, participants in subsample 1 had had limb loss for more years (mn 33.7 years, sd 18.6) than did those in subsample 2 (24.4 years, sd 19.6). The full sample included 113 (19.7%) women, most of whom were in subsample 2. The most common amputation levels were transradial (55.2%), transhumeral (29%), and shoulder (9.3%), followed by bilateral amputation (6.2%). All 42 participants with bilateral amputation were in subsample 2. There were 475 (70.2%) prosthesis users in the full sample.

### Factor analysis

Preliminary EFA of all 46 PEM items identified 10 eigenvalues >1 and resulted in a seven-factor solution. Because items related to task performance and handling fragile/delicate objects items loaded well on two factors, these were combined into a single factor that we labeled self-

**Table 2. Characteristics of the field study sample.**

| ' | Sub-sample 1 (N = 320) | Sub-sample 2 (N = 357) | Full Sample (N = 677) | Full Sample | | Test-retest Sample (N = 50) |
|---|---|---|---|---|---|---|
| | | | | Prosthesis Users (N = 475) | Nonusers (N = 202) | |
| | Mn (sd) | Mn (sd) | Mn (sd) | Mn (sd) | Mn (sd) | Mn (sd) |
| **Age** | 63.8 (13.2) | 59.0 (15.5) | 61.3 (14.6) | 61.5 (14.8) | 61.0 (14.2) | 61.1 (14.2) |
| **Years since amputation** | 33.7 (18.6) | 24.4 (19.6) | 29.1 (19.6) | 29.6 (20.1) | 27.9 (18.6) | 31.7 (19.7) |
| | N (%) | N (%) | N (%) | N (%) | N (%) | N (%) |
| **Status** | | | | | | |
| Veteran | 307 (95.9) | 212 (60.4) | 519 (77.4) | 362 (77.0) | 157 (78.1) | 47 (94.0) |
| Civilian | 13 (4.1) | 138 (39.3) | 151 (22.5) | 108 (23.0) | 43 (21.4) | 3 (6.0) |
| Unknown | 0 (0.0) | 1 (0.3) | 1 (0.2) | 0 (0.0) | 1 (0.5) | 0 (0.0) |
| **Gender** | | | | | | |
| Female | 9 (2.8) | 124 (34.7) | 133 (19.7) | 95 (20.0) | 38 (18.8) | 2 (4.0) |
| Male | 311 (97.2) | 233 (65.3) | 544 (80.4) | 380 (80.0) | 164 (81.2) | 48 (96.0) |
| **Race** | | | | | | |
| White | 250 (81.3) | 294 (82.4) | 554 (81.8) | 390 (82.1) | 164 (81.2) | 40 (80.0) |
| Black | 31 (9.7) | 30 (8.4) | 61 (9.0) | 42 (8.8) | 19 (9.4) | 2 (4.0) |
| Unknown | 18 (5.6) | 23 (6.4) | 41 (6.1) | 25 (5.3) | 16 (7.9) | 6 (12.0) |
| Mixed | 11 (3.4) | 10 (2.8) | 21 (3.1) | 18 (3.8) | 3 (1.5) | 2 (4.0) |
| **Amputation level** | | | | | | |
| Shoulder | 35 (10.9) | 28 (7.8) | 63 (9.3) | 27 (5.7) | 36 (17.8) | 10 (20.0) |
| Transhumeral | 109 (34.1) | 89 (24.9) | 198 (29.3) | 103 (21.7) | 95 (47.0) | 15 (30.0) |
| Transradial | 176 (55.0) | 198 (55.5) | 374 (55.2) | 309 (65.1) | 65 (32.2) | 15 (30.0) |
| Bilateral | 0 (0.0) | 42 (11.8) | 42 (6.2) | 36 (7.6) | 6 (3.0) | 10 (20.0) |
| **Amputation etiology** | | | | | | |
| Combat | 98 (30.6) | 55 (18.1) | 153 (24.5) | 121 (28.1) | 32 (16.6) | 16 (32.0) |
| Accident | 204 (63.8) | 191 (62.8) | 395 (63.3) | 264 (61.3) | 131 (67.9) | 31 (62.0) |
| Burn | 30 (9.4) | 35 (11.5) | 65 (10.4) | 52 (12.1) | 13 (6.7) | 9 (18.0) |
| Cancer | 13 (4.1) | 23 (7.6) | 36 (5.8) | 19 (4.4) | 17 (8.8) | 3 (6.0) |
| Diabetes | 1 (0.3) | 1 (0.3) | 2 (0.3) | 2 (.5) | 0 (0.0) | 0 (0.0) |
| Infection | 25 (7.8) | 2 (16.5) | 75 (12.0) | 53 (12.3) | 22 (11.4) | 6 (12.0) |
| Congenital | 0 (0.0) | 53 (14.9) | 269 (43.2) | 182 (42.3) | 87 (45.1) | 0 (0.0) |
| Other | 115 (35.9) | 154 (50.8) | 53 (7.6) | 44 (9.3) | 9 (4.5) | 16 (32.0) |
| **Current prosthesis user** | | | | | | |
| Yes | 109 (34.1) | 93 (26.1) | 475 (70.2) | 475 (100.0) | 0 (0.0) | 50 (100.0) |
| **Primary* aprosthesis type** | | | | | | |
| Body-powered | 155 (73.5) | 158 (59.9) | 313 (65.9) | 313 (65.9) | NA | 41 (82.0) |
| Myoelectric | 44 (20.9) | 72 (27.3) | 116 (24.4) | 116 (24.4) | NA | 6 (12.0) |
| Hybrid | 0 (0.0) | 4 (1.5) | 4 (0.8) | 4 (0.8) | NA | 0 (0.0) |
| Cosmetic | 8 (3.8) | 20 (7.6) | 28 (5.9) | 28 (5.9) | NA | 2 (4.0) |
| Sport | 4 (1.9) | 6 (2.3) | 10 (2.1) | 10 (2.1) | NA | 1 (2.0) |
| Unknown | 0 (0.0) | 4 (1.5) | 4 (0.8) | 4 (0.8) | NA | 0 (0.0) |

*Primary type of prosthesis is the prosthesis type used most often. Some participants used more than one type of prosthesis.

efficacy. The resulting six scales were labeled based on an evaluation of item content as: social interaction, self-efficacy, embodiment, intuitiveness, wellbeing, and self-consciousness. Cronbach alpha for the six factors ranged from 0.78 to 0.95 in subsample 1.

CFA of the six proposed scales in subsample 1 had acceptable fit indices (CFI = 0.949, TLI = 0.946, RMSEA = 0.042). CFA in subsample 2 also had acceptable fit indices (CFI = 0.930, TLI = 0.926, RMSEA = 0.053) and TCC was >0.97 for each scale. All factors were considered unidimensional; however, both the self-efficacy and intuitiveness factors had high RMSEA. After dropping items with high residual correlations (Table 3) RMSEA

**Table 3. Original PEM items, items utilized in field testing, items not retained and items in the final modified PEM.**

| Original Items | Items used in Field Testing | Final PEM Subscale |
|---|---|---|
| Hold someone else's hand while walking without hurting them | **Using your prosthesis to grasp someone else's hand while walking without hurting them** | Social Interaction |
| Hold someone else's hand while walking without hurting them | *Opening your terminal device when shaking hands* | Social Interaction |
| I am comfortable using my prosthesis to hold hands with someone close to me. | *Grasping with your prosthesis to shake hands with someone close to you* | Social Interaction |
| I am comfortable using my prosthesis to shake hands with someone I just met. | *Grasping with your prosthesis to shake hands with someone you just met* | Social Interaction |
| I am comfortable using my prosthesis to shake hands with someone I know well | *Grasping with your prosthesis to shake hands with someone you know well* | Social Interaction |
| I can use my prosthesis to gently squeeze someone else's hand | **Using your prosthesis to gently squeeze someone else's hand** | Social Interaction |
| I would use my prosthesis when embracing someone I cared about. | **Using your prosthesis when embracing someone you care about** | Social Interaction |
| I can convey a friendly or caring touch using my prosthesis. | **Using your prosthesis to convey a friendly or caring touch** | Social Interaction |
| I can use my prosthesis to gently pat a dog or cat | **Using your prosthesis to gently pat a dog or cat** | Social Interaction |
| I can control whether I deliver a soft or a firm touch when patting someone on the back using my prosthesis. | **Using your prosthesis to deliver a soft or a firm touch when patting someone on the back** | Social Interaction |
| Wearing my prosthesis interferes with my physical and intimate relationships | *Using your prosthesis in your physical and intimate relationships* | Social Interaction |
| Carry a small object (such as a coin) without dropping it | **Using your prosthesis to carry a small object, such as a coin, without dropping it** | Self-efficacy |
| Pick up an open plastic water bottle without dropping or crushing it | **Using your prosthesis to pick up an open plastic water bottle without dropping or crushing it** | Self-efficacy |
| — | Holding a dinner glass using your prosthesis | Self-efficacy |
| — | Tying a knot using your prosthesis | Self-efficacy |
| Carry a slippery object (such as a silk scarf or tie) without dropping it. | **Using your prosthesis to carry a slippery object, such as a silk scarf or tie, without dropping it** | Self-efficacy |
| — | Using your prosthesis to carry a laundry basket | Self-efficacy |
| I was willing to try new tasks with my prosthesis. | **Trying new tasks with your prosthesis** | Self-efficacy |
| — | Using your prosthesis to eat with a knife and fork while in a restaurant | Self-efficacy |
| Drink from a paper cup without dropping or crushing it | **Using your prosthesis to drink from a paper cup without dropping or crushing it** | Self-efficacy |
| Pick up a Ritz cracker without breaking it | **Using your prosthesis to pick up a Ritz cracker without breaking it** | Self-efficacy |
| — | Using your prosthesis to pick up fragile objects | Self-efficacy |
| — | Using your prosthesis to hold a child | Self-efficacy |
| — | Using your prosthesis to pick up a small child* | |
| My prosthesis is a part of me | **My prosthesis is a part of me** | Embodiment |
| I feel more complete when wearing my prosthesis | **I feel more complete when wearing my prosthesis** | Embodiment |
| My prosthesis felt like it was my hand | *My prosthesis feels like a hand* | Embodiment |
| My prosthesis is an extension of me | *My prosthesis is an extension of my body* | Embodiment |
| I use my prosthesis to express myself | **I use my prosthesis to express myself** | Embodiment |
| When I remove my prosthesis, I feel. . . a sense of loss | *When I take off my prosthesis, I feel a sense of loss**** | |
| I avoided using my prosthesis to do things because it slowed me down. | *Using my prosthesis slows me down* | Intuitiveness |

*(Continued)*

**Table 3.** (Continued)

| Original Items | Items used in Field Testing | Final PEM Subscale |
|---|---|---|
| — | Using my prosthesis requires concentration | Intuitiveness |
| Using my prosthesis required a lot of focus | *Using my prosthesis requires visual focus** | |
| — | Using my prosthesis is not natural | Intuitiveness |
| — | Using my prosthesis is clumsy | Intuitiveness |
| I looked forward to removing my prosthesis | *I look forward to removing my prosthesis so I can be more comfortable*** | |
| When I remove my prosthesis, I feel more confident | *I feel confident (without a prosthesis)* | Wellbeing |
| When I remove my prosthesis, I feel less confident | | |
| When I remove my prosthesis, I feel. . . Happy | **I feel happy (without a prosthesis)** | Wellbeing |
| When I remove my prosthesis, I feel more whole | *I feel whole (without a prosthesis)* | Wellbeing |
| When I remove my prosthesis, I feel less whole | | |
| When I remove my prosthesis, I feel. . .Relieved | **I feel relieved (without a prosthesis)** | Wellbeing |
| — | I feel relaxed (without a prosthesis) | Wellbeing |
| When I remove my prosthesis, I feel. . .Free | **I feel free (without a prosthesis)** | Wellbeing |
| — | I feel vulnerable (without a prosthesis) | Self-consciousness |
| — | I feel incomplete (without a prosthesis) | Self-consciousness |
| When I remove my prosthesis, I feel. . .Different from others | **I feel different from others (without a prosthesis)** | Self-consciousness |
| When I remove my prosthesis, I feel more shy | *I feel shy in public (without a prosthesis)* | Self-consciousness |
| When I remove my prosthesis, I feel less shy | | |

^ **Bold** items were in original PEM verbatim; *Italicized* items were addressed in an original item but wording was revised; plain text are new items.

* Dropped in CFA–negative residual correlation and highly correlated with other items.

** Dropped in Rasch analysis–infit>1.4.

improved from 0.173 to 0.115 for the self-efficacy factor and from 0.215 to 0.083 for the intuitiveness factor. There was no residual correlation greater than 0.2 for all other factors.

## Rasch analysis

**Monotonicity.** In the social interaction, self-efficacy, embodiment, and self-consciousness scales, category characteristic curves revealed disordered threshold parameters in the middle three response categories for all items. To address this, the middle three categories were merged resulting in three-category responses. In the intuitiveness and wellbeing scales, disordered threshold parameters occurred only in the middle and second highest response categories for all items, so only these categories were combined resulting in four-category responses.

**Fit.** After collapsing response categories, the partial credit model identified misfit items in each scale (Table 3). In the self-efficacy, intuitiveness and self-consciousness scales, all infit and outfit values were <1.4. Two items were dropped due to poor fit: one in the embodiment scale ('when I take off my prosthesis, I feel a sense of loss', which had infit = 1.49) and one in the wellbeing scale ('I look forward to removing my prosthesis so I can be more comfortable', which had infit = 1.88). In the social interaction scale, all infit values were <1.4, except the item 'using your prosthesis in your physical and intimate relationships', which had infit = 1.41. This item was retained given the borderline acceptable item fit and importance of the item content.

In the social interaction, self-efficacy, embodiment, intuitiveness, wellbeing and self-consciousness scales, the percent of variance explained by the model was 72.4%, 72.3%, 78.0%, 57.8%, 65.1% and 51.6%, respectively, and the eigenvalues of the first contrasts were 1.9, 2.2, 1.4, 1.5, 1.6, and 1.6, respectively. While the self-efficacy eigenvalue is slightly higher than the criteria of 2, a large proportion of the observed variance was explained by the Rasch model. Therefore, we considered the unidimensionality of the scale acceptable.

For all PEM scales, no positive residual correlations were greater than 0.4. One item pair ('using my prosthesis is natural' and 'using my prosthesis requires concentration') in the intuitiveness scale had a high negative residual correlation of -0.47. Given that there is no clear conceptual explanation for higher negative residual correlations and our belief that these items targeted different aspects of intuitiveness, we retained both items. The items utilized in field testing, items dropped, and items in the final modified PEM scales are shown in Table 3.

**DIF.** All items with slight to moderate or moderate to severe DIF (as confirmed by Lasso methods) are detailed in Table 4 along with the directionality of the DIF. For all scales with DIF items, Rasch item calibration was reevaluated in final partial credit models shown in Table 5.

**Cross-validation.** Cross validation techniques confirmed that there was little difference between the partial credit model and generalized partial credit models in estimating item responses. The difference of bias (or RMSD) across models and scales was less than 0.1, which confirmed that the partial credit model was acceptable for these data.

## Final scales

The final scales are provided in S3 Appendix. After removal of 2 items during Rasch analyses, the overall model fit remained acceptable CFI = 0.973, TLI = 0.979, RMSEA = 0.038. Factor correlations are reported in Table 6. The highest correlation was between the self-efficacy and social interaction scales (0.719, P<0.05). Tables 1–6 in S4 Appendix provide information for conversion of scale summary totals to T-scores, with separate scoring tables for scales with DIF items.

**Table 4. Differential Item Functioning (DIF) results.**

| Scale/Item | DIF contrast | Joint SE | DIF Severity | DIF by | DIF directionality: More difficult for... |
|---|---|---|---|---|---|
| **Social Interaction** | | | | | |
| Opening your terminal device when shaking hands | -0.70 | 0.24 | * | Age | Those >65 |
| **Self-efficacy** | | | | | |
| Using your prosthesis to hold a child | 0.72 | 0.22 | * | Gender | Men |
| Tying a knot using your prosthesis | -1.93 | 0.35 | ** | Laterality | Bilateral amputation |
| Using your prosthesis to drink from a paper cup without dropping or crushing it | 2.20 | 0.34 | ** | Laterality | Unilateral amputation |
| **Embodiment** | | | | | |
| I use my prosthesis to express myself | -0.94 | 0.22 | ** | Age | Those >65 |
| **Intuitiveness (No DIF items)** | | | | | |
| **Wellbeing** | | | | | |
| I feel relaxed (without a prosthesis) | -0.78 | 0.13 | * | Prosthesis use | Nonusers |
| **Self-consciousness (No DIF items)** | | | | | |

*Slight to moderate: DIF contrast >0.43 and >2*SE (Standard Error).

**Moderate to severe: DIF contrast>0.64 and >0.43+2*SE.

**Table 5. Partial credit model of PEM subscales.**

| | Logit Model | | T-Score Model | | Infit | | Outfit | |
|---|---|---|---|---|---|---|---|---|
| | Measure | SE | Measure | SE | MNSQ | ZSTD | MNSQ | ZSTD |
| **Social Interaction (N = 401)** | | | | | | | | |
| Grasping with your prosthesis to shake hands with someone you just met | 1.01 | 0.12 | 56.36 | 0.63 | 0.65 | -4.6 | 0.57 | -3.8 |
| Using your prosthesis in your physical and intimate relationships | 0.81 | 0.12 | 55.31 | 0.63 | 1.40 | 4.2 | 1.52 | 3.3 |
| Using your prosthesis to gently squeeze someone else's hand | 0.57 | 0.12 | 54.05 | 0.63 | 0.82 | -2.3 | 0.82 | -1.6 |
| Opening your terminal device when shaking hands (>65) | 0.36 | 0.19 | 52.94 | 1.00 | 1.38 | 2.5 | 1.41 | 1.9 |
| Grasping with your prosthesis to shake hands with someone close to you | 0.27 | 0.11 | 52.47 | 0.58 | 0.72 | -3.6 | 0.71 | -2.7 |
| Using your prosthesis to grasp someone else's hand while walking without hurting them | 0.03 | 0.11 | 51.21 | 0.58 | 1.38 | 4.3 | 1.56 | 5.3 |
| Using your prosthesis to deliver a soft or a firm touch when patting someone on the back | -0.05 | 0.12 | 50.79 | 0.63 | 0.82 | -2.3 | 0.78 | -2.7 |
| Using your prosthesis to gently pat a dog or cat | -0.25 | 0.11 | 49.74 | 0.58 | 1.08 | 1.0 | 1.10 | 1.0 |
| Grasping with your prosthesis to shake hands with someone you know well | -0.37 | 0.11 | 49.11 | 0.58 | 0.70 | -3.8 | 0.70 | -2.7 |
| Using your prosthesis to convey a friendly or caring touch | -0.40 | 0.12 | 48.95 | 0.63 | 1.02 | 0.2 | 1.01 | 0.1 |
| Opening your terminal device when shaking hands (≤65) | -0.43 | 0.16 | 48.80 | 0.84 | 1.16 | 1.4 | 1.17 | 1.4 |
| Using your prosthesis when embracing someone you care about | -1.55 | 0.11 | 42.92 | 0.58 | 1.11 | 1.4 | 1.15 | 1.2 |
| **Self-efficacy (N = 417)** | | | | | | | | |
| Using your prosthesis to drink from a paper cup without dropping or crushing it (Unilateral) | 1.57 | 0.11 | 52.74 | 0.41 | 1.09 | 1.1 | 0.98 | -0.1 |
| Tying a knot using your prosthesis (Bilateral) | 1.56 | 0.35 | 52.70 | 1.30 | 1.12 | 0.6 | 1.01 | 0.1 |
| Using your prosthesis to pick up a Ritz cracker without breaking it | 1.18 | 0.10 | 51.29 | 0.37 | 0.94 | -0.8 | 0.84 | -1.5 |
| Using your prosthesis to pick up fragile objects | 0.83 | 0.10 | 49.98 | 0.37 | 0.87 | -1.9 | 0.84 | -2.1 |
| Holding a dinner glass using your prosthesis | 0.64 | 0.10 | 49.28 | 0.37 | 0.85 | -2.2 | 0.92 | -0.7 |
| Using your prosthesis to hold a child (Men) | 0.61 | 0.11 | 49.16 | 0.41 | 1.12 | 1.4 | 1.09 | 0.7 |
| Using your prosthesis to pick up an open plastic water bottle without dropping or crushing it | 0.21 | 0.10 | 47.68 | 0.37 | 0.91 | -1.4 | 0.93 | -0.9 |
| Using your prosthesis to eat with a knife and fork while in a restaurant | -0.06 | 0.09 | 46.67 | 0.33 | 0.88 | -1.8 | 0.84 | -1.9 |
| Using your prosthesis to hold a child (Women) | -0.18 | 0.21 | 46.22 | 0.78 | 1.08 | 0.5 | 1.08 | 0.6 |
| Tying a knot using your prosthesis (Unilateral) | -0.47 | 0.10 | 45.15 | 0.37 | 1.06 | 0.9 | 1.03 | 0.4 |
| Using your prosthesis to drink from a paper cup without dropping or crushing it (Bilateral) | -0.81 | 0.33 | 43.88 | 1.23 | 1.01 | 0.1 | 0.82 | -0.3 |
| Using your prosthesis to carry a slippery object, such as a silk scarf or tie, without dropping it | -0.99 | 0.09 | 43.21 | 0.33 | 1.04 | 0.6 | 1.04 | 0.4 |
| Using your prosthesis to carry a small object, such as a coin, without dropping it | -1.06 | 0.09 | 42.95 | 0.33 | 1.16 | 2.2 | 1.13 | 1.2 |
| Trying new tasks with your prosthesis | -1.21 | 0.11 | 42.39 | 0.41 | 1.05 | 0.8 | 1.06 | 0.7 |
| Using your prosthesis to carry a laundry basket | -1.81 | 0.10 | 40.16 | 0.37 | 1.14 | 1.8 | 1.10 | 0.8 |
| **Embodiment (N = 418)** | | | | | | | | |
| I use my prosthesis to express myself (>65 yrs) | 1.67 | 0.16 | 56.96 | 0.91 | 1.38 | 3.2 | 1.32 | 1.5 |
| My prosthesis feels like a hand | 1.62 | 0.11 | 56.67 | 0.63 | 1.14 | 2.0 | 1.06 | 0.6 |
| I use my prosthesis to express myself (≤65 yrs) | 0.45 | 0.15 | 49.98 | 0.86 | 1.15 | 1.5 | 1.15 | 1.4 |
| I feel more complete when wearing my prosthesis | -1.08 | 0.12 | 41.24 | 0.69 | 0.91 | -1.2 | 0.86 | -1.3 |
| My prosthesis is an extension of my body | -1.11 | 0.12 | 41.07 | 0.69 | 0.78 | -3.0 | 0.71 | -3.1 |
| My prosthesis is a part of me | -1.55 | 0.12 | 38.55 | 0.69 | 0.93 | -1.0 | 0.83 | -1.5 |
| **Intuitiveness (N = 418)** | | | | | | | | |
| Using my prosthesis requires concentration | 0.53 | 0.07 | 50.44 | 0.40 | 1.08 | 1.2 | 1.12 | 1.6 |
| Using my prosthesis is not natural | 0.03 | 0.07 | 47.58 | 0.40 | 1.03 | 0.5 | 1.07 | 0.9 |
| Using my prosthesis is clumsy | -0.04 | 0.08 | 47.18 | 0.46 | 0.86 | -2.1 | 0.86 | -2.0 |
| Using my prosthesis slows me down | -0.52 | 0.07 | 44.44 | 0.40 | 1.03 | 0.4 | 1.03 | 0.4 |
| **Wellbeing (N = 676)** | | | | | | | | |
| I feel relieved (without a prosthesis) | 0.86 | 0.06 | 48.74 | 0.31 | 1.08 | 1.4 | 1.07 | 1.2 |
| I feel relaxed (without a prosthesis) (Users) | 0.36 | 0.12 | 46.14 | 0.62 | 0.91 | -0.8 | 0.93 | -0.6 |
| I feel whole (without a prosthesis) | 0.07 | 0.06 | 44.63 | 0.31 | 0.96 | -0.7 | 0.97 | -0.4 |
| I feel free (without a prosthesis) | 0.04 | 0.06 | 44.47 | 0.31 | 1.00 | 0.0 | 0.99 | -0.1 |

*(Continued)*

**Table 5.** (Continued)

| | Logit Model | | T-Score Model | | Infit | | Outfit | |
|---|---|---|---|---|---|---|---|---|
| | Measure | SE | Measure | SE | MNSQ | ZSTD | MNSQ | ZSTD |
| I feel confident (without a prosthesis) | -0.28 | 0.06 | 42.81 | 0.31 | 1.24 | 3.7 | 1.34 | 4.1 |
| I feel relaxed (without a prosthesis) *(Nonusers)* | -0.46 | 0.07 | 41.88 | 0.36 | 0.82 | -2.6 | 0.78 | -2.6 |
| I feel happy (without a prosthesis) | -0.59 | 0.06 | 41.20 | 0.31 | 0.88 | -2.0 | 0.86 | -1.9 |
| **Self-consciousness (N = 668)** | | | | | | | | |
| I feel different from others (without a prosthesis) | 0.91 | 0.08 | 51.78 | 0.36 | 1.05 | 0.9 | 1.06 | 1.0 |
| I feel vulnerable (without a prosthesis) | -0.12 | 0.08 | 47.13 | 0.36 | 1.06 | 1.1 | 1.06 | 1.1 |
| I feel incomplete (without a prosthesis) | -0.24 | 0.09 | 46.59 | 0.41 | 0.92 | -1.4 | 0.91 | -1.6 |
| I feel shy in public (without a prosthesis) | -0.55 | 0.08 | 45.19 | 0.36 | 0.97 | -0.4 | 0.95 | -0.8 |

*SE: standard error; MNSQ: mean squared; ZSTD: z-standardized.

## Reliability

**Item-person maps.** Rasch item-person maps for each scale (Fig 1) showed that item difficulties (including lowest and highest categories) sufficiently covered the range of person ability scores for the social interaction, self-efficacy and embodiment scales. The grey bar in the figures indicates the score range where reliability was >0.8. There was clustering at the higher levels of ability for the wellbeing and self-consciousness scales, suggesting a need for more difficult items to improve coverage of person ability.

**Rasch reliability.** For each of the scales, person reliability was 0.81 for social interaction, 0.80 for self-efficacy, 0.77 for embodiment, 0.72 for wellbeing, 0.66 for intuitiveness, and 0.65 for self-consciousness. Cronbach's alpha was 0.90 for social interaction, 0.80 for self-efficacy, and ranged from 0.71 to 0.79 for all other scales (see Table 7).

**Test-retest reliability.** ICCs(3,1) were 0.82, 0.85, 0.74, 0.55, 0.48, 0.63 for the social interaction, self-efficacy, embodiment, intuitiveness, wellbeing, and self-consciousness scales, respectively. Confidence intervals are shown in Table 7. MDC90 and 95% values are also shown in Table 7.

**Floor and ceiling.** For all six PEM scales, less than 15% of the sample were at the floor (1.9%-11.5%) of possible item responses. However, ceiling effects were observed for the self-consciousness (15.8% at ceiling) and wellbeing (16.4% at ceiling) scales.

## Discussion

While prior work utilized an earlier prototype of the PEM to study the psychosocial experience of persons with sensory enabled prostheses, this is the first study that examined the

**Table 6. Factor correlations in final CFA (N = 677).**

| | Social Interaction | Self-efficacy | Embodiment | Intuitiveness | Wellbeing | Self-consciousness |
|---|---|---|---|---|---|---|
| Social Interaction | 1 | | | | | |
| Self-Efficacy | 0.719* | 1 | | | | |
| Embodiment | 0.478* | 0.428* | 1 | | | |
| Intuitiveness | 0.242* | 0.338* | 0.561* | 1 | | |
| Wellbeing | 0.049 | 0.018 | -0.179* | -0.034 | 1 | |
| Self-consciousness | 0.054 | 0.052 | -0.175* | 0.218* | 0.489* | 1 |

*significant at p<0.05.

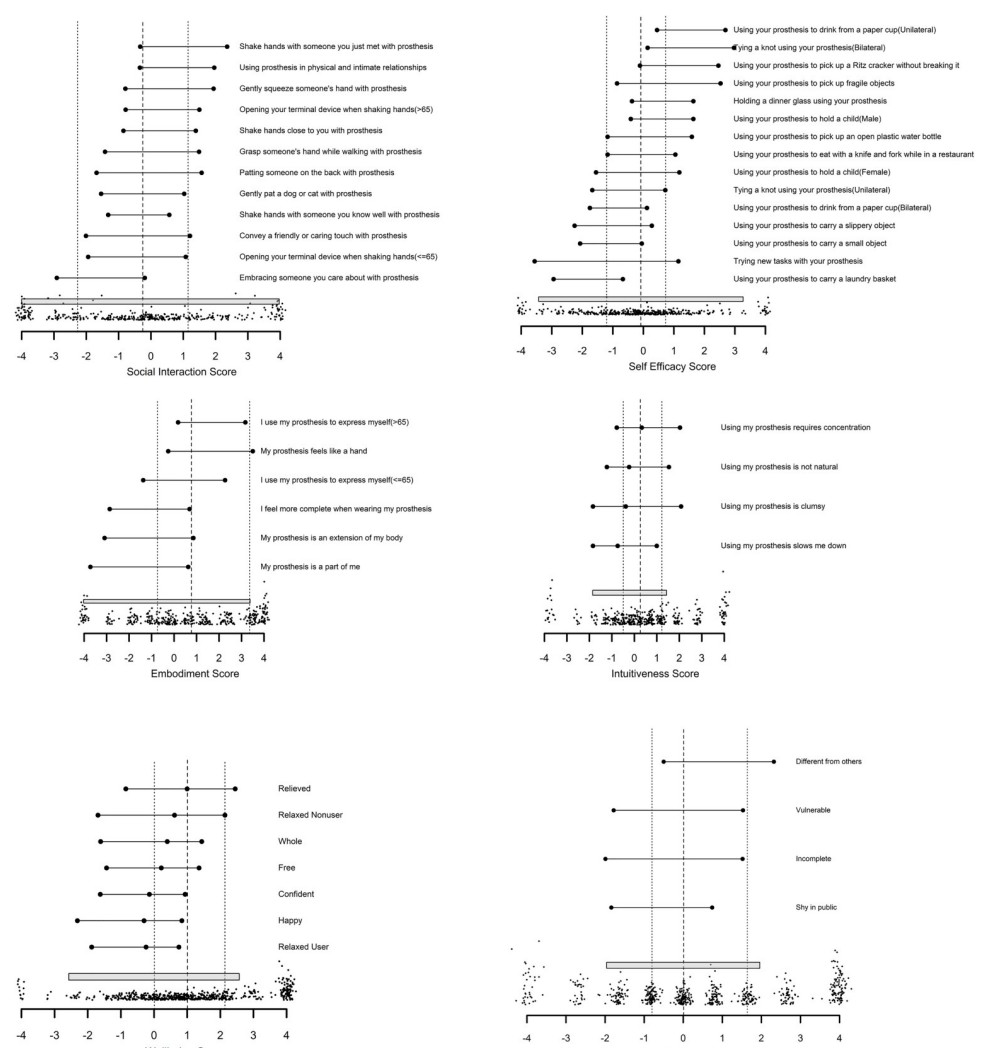

**Fig 1. PEM item maps.** 1a) Social Touch, 1b) Self-efficacy, 1c) Embodiment, 1d) Intuitiveness, 1e) Wellbeing, 1f) Self-consciousness. Level of person ability with 50% probability of selecting each category (vs any higher category) for each item is shown on the top of each panel; a histogram of person scores is shown on the bottom. Dotted vertical lines show 25th and 75th percentiles; dashed line is median score. Grey bar indicates the score range with reliability >0.8.

psychometric properties of the PEM. Our analyses yielded a refined set of scales that can be utilized in upper limb prosthetics research. The novel PEM scales quantify key psychosocial aspects of the prosthesis experience that we believe are important for understanding the

**Table 7. PEM subscale ICCs, MDCs, person reliability, Cronbach alphas, and floor and ceiling effects.**

|  | Test-Retest Sample | | | | Full Sample | | | |
|---|---|---|---|---|---|---|---|---|
|  | N | ICC (95% CI) | MDC 90 | MDC 95 | N | Person Reliability (Rasch) | Cronbach Alpha | N (%) at Floor | N (%) at Ceiling |
| **Social Interaction** | 42 | 0.82 (0.69, 0.90) | 13.6 | 11.4 | 401 | 0.81 | 0.90 | 46 (11.5) | 36 (9.9) |
| **Self-Efficacy** | 49 | 0.85 (0.75, 0.91) | 12.8 | 10.8 | 417 | 0.80 | 0.80 | 13 (3.1) | 16 (3.8) |
| **Embodiment** | 50 | 0.74 (0.59, 0.85) | 14.0 | 11.7 | 418 | 0.77 | 0.71 | 15 (3.6) | 46 (11.0) |
| **Intuitiveness** | 50 | 0.55 (0.32, 0.72) | 18.2 | 15.3 | 418 | 0.66 | 0.73 | 17 (4.1) | 33 (7.9) |
| **Wellbeing** | 49 | 0.48 (0.23, 0.67) | 19.2 | 16.1 | 676 | 0.72 | 0.79 | 13 (1.9) | 110 (16.4) |
| **Self-consciousness** | 49 | 0.63 (0.43, 0.78) | 16.5 | 13.8 | 668 | 0.65 | 0.77 | 36 (5.4) | 106 (15.8) |

impact of advanced prosthesis technologies, including neuroprostheses. Although measures and functional tests have previously been devised to address some aspects of the psychosocial constructs covered in the PEM such as embodiment [24, 25], self-efficacy [26], and social burden [5], few of these measures, if any, have undergone rigorous empirical testing to examine structural validity and reliability specifically in persons with upper limb loss. Thus, our work addresses an important gap in measuring psychosocial experiences in persons using upper limb prostheses. The strength of our work is the large sample size and use of contemporary measurement methods that allowed us to identify unidimensional constructs, differential item functioning by participant characteristics, and the ordering of items within constructs.

We identified six scales from our analyses: social interaction, self-efficacy, embodiment, intuitiveness, wellbeing, and self-consciousness. CFA and Rasch analyses of the revised PEM showed that all scales were unidimensional. Three scales–social interaction, self-efficacy, and embodiment–had strong person reliability, internal consistency, and test-retest reliability. Two scales–intuitiveness and self-consciousness–had marginal person reliability, fair internal consistency, but questionable test-rest reliability. The wellbeing scale had fair person reliability and internal consistency, but poor test-retest reliability. Thus, we conclude that the PEM social interaction, self-efficacy, and embodiment scales are reliable and valid and recommend that they be implemented in prosthetics research. The other three scales should be used cautiously, given that additional refinement may be needed to improve reliability. In particular, future work should focus on increasing the number of items and difficulty range of the intuitiveness scale.

## Difficulty of items within each scale

One of the strengths of our approach was that Rasch item calibration rank orders the difficulty of scale items, as well as determining differential functioning of items by demographic groups. For the self-efficacy scale, tasks requiring fine control of grip force, such as drinking from a paper cup without dropping or crushing it, were more difficult than tasks in which grip force regulation was not as important, such as carrying a laundry basket. The DIF analyses suggested that bilateral amputees found tasks requiring fine force regulation with the prosthesis easier than unilateral amputees, but certain dexterous bilateral activities, such as tying a knot, more difficult than unilateral amputees. Neither of these results are surprising, given that bilateral amputees cannot rely on their intact limb for fine force regulation and thus develop more dexterity on their dominant prosthetic side [27]. Further, bilateral amputees cannot use an intact limb during bilateral activities, making these activities more difficult. Although we adjusted for items with moderate to severe DIF in our scoring, we note that this adjustment had little effect (< 1.5 point) on the average scores once transformed to T score. Nevertheless, until further research confirms or refutes our findings, we recommend adjustment for DIF.

For the social interaction scale, difficulty of the items appears to depend on the relationship of the prosthesis user to the person with whom they are interacting. Items related to interactions with strangers ('grasping with your prosthesis to shake hands with someone you just met') were more difficult than items related to interactions with close friends and family ('grasping with your prosthesis to shake hands with someone you know well'). In addition, the degree to which control of the terminal device was required for completion of a task was related to the difficulty of the item within the social interaction scale. Social interactions involving closing the terminal device, such as shaking or holding hands, were more difficult than social interactions in which grasp is not required, such as 'using your prosthesis when embracing someone you care about.'

The item difficulty ordering in the embodiment scale confirms the previously hypothesized hierarchy of prosthesis embodiment. The item 'my prosthesis is a part of me' had the lowest

difficulty, while the item 'my prosthesis feels like a hand' was at the hardest end of the spectrum. This spectrum of embodiment has been previously theorized by researchers describing differences in extensions of the body for tool use and incorporation of prostheses to replace lost limbs [28, 29]. In addition, item difficulty revealed differences between the concepts of agency and ownership, which are thought to be aspects of embodiment. The item 'I use my prosthesis to express myself' likely pertains to the concept of agency. It implies a high level of motor control of the device so that one's expressions are transferred to the device, and the control and authorship of an action is a core concept in feelings of agency [24, 30]. This self-expression item was found to be more difficult than the feeling of ownership over the device as described by the item 'my prosthesis is a part of me,' indicating that the feeling of ownership may be more prevalent in prosthetic users while the feeling of agency is more difficult to achieve. This difference in difficulty across ownership and agency is curious since the feeling of ownership requires multi-sensory integration [31, 32], and commercial prosthetic devices do not provide sensory feedback from the terminal device. On the other hand, the prosthetic control available today may not be sufficient to enable robust self-expression and thereby reduces the feeling of agency over the device. Interestingly, DIF analysis showed that this same item ('I use my prosthesis to express myself') was more difficult for respondents over 65. This finding suggests that agency becomes more difficult with increasing age, or that this cohort is less likely to try to express themselves through gesture involving the prosthesis. However, this observation may be confounded by prosthesis type, given that older prosthesis users are more likely to use body-powered devices, and/or less likely to have anthropomorphic terminal devices. Additional research is needed to understand how responses to this item vary by prosthesis type. Nonetheless, the ability to rank order the item difficulty provides insight into the breadth of each construct and highlights the range of the psychosocial experience of upper limb prosthesis users.

### Challenges with wellbeing and self-consciousness scales

In the original PEM, a scale called body image included many of the items that were separated in this analysis into two distinct scales: wellbeing and self-consciousness. In our analyses, these scales were only moderately correlated with each other, suggesting that the content represented two separate constructs. Content evaluation suggests that the self-consciousness scale addresses social pressures or social anxiety, while the wellbeing scale addresses self-image and body acceptance. The instructions for the well-being and self-consciousness items were slightly different for prosthesis users and non-users. Whereas prosthesis users were asked to report on their experiences without wearing a prosthesis in the prior four weeks, non-users were asked to rate their experience in the prior four weeks with no mention of prosthesis use. We observed DIF between users and non-users for the item 'I feel relaxed'–suggesting that prosthesis users interpreted the item as a contrast between wearing vs not wearing a prosthesis (as intended), while non-users interpreted this item as pertaining to their general mood, given that they could not contrast their experiences when wearing a prosthesis. Given these differences, and the existence of other validated generic measures of emotional wellbeing [33], the unique contribution of the wellbeing scale is diminished. In addition, until further work is done to examine concurrent validity of these scales, we recommend that the PEM be administered for prosthesis users only.

### Correlation between scales

It was initially surprising that embodiment and intuitiveness were moderately correlated. This relationship may be explained by the fact that intuitiveness of prosthesis use could be related

to the feeling of agency, a component of embodiment. Given, the lower reliability scores of the intuitiveness scale, we considered merging the items from the intuitiveness and embodiment scales, but after exploratory CFA found that they did not fit well. In other words, the items within the intuitiveness and embodiment scales are different, but related constructs.

We identified a strong correlation between the social interaction and self-efficacy scales (r = 0.72) indicating that confidence is key to using a prosthesis in social interaction. Because both scales had high reliability, we did not attempt to merge them into a single scale.

### The value of contemporary measurement methods in prosthetics research

The psychometric assessment of aggregate responses of prosthesis users provides a foundational understanding of hard-to-measure, latent, psychosocial concepts related to prosthesis use. The experience of prosthesis embodiment, for example, is associated with successful prosthesis use [34], but definitions of embodiment vary widely in the literature and controversy exists about the sub-categories within embodiment [35–37]. For example, Schofield et al. detailed the mechanism of embodiment as being the combined experience of owning and controlling a body and its parts [36]. Zbinden et al. compiled definitions for embodiment, ownership, and agency and reviewed explicit and implicit outcome measures to quantify these psychological phenomena [35]. These recent publications highlight that (a) there are no suitable tools currently available to measure prosthetic embodiment and (b) the lack of measurement tools limits research in this area.

In our study, we were able to utilize the results of factor and Rasch analyses to confirm unidimensionality of our scales, which suggests that the items in each unidimensional scale measure a singular, latent construct. This empirically driven method of defining latent constructs relevant to prosthesis use contrasts with other approaches where experimenters derive definitions based upon observational or qualitative data. In future work, new item sets specifically related to ambiguous or controversial latent constructs of prosthesis use (such as ownership, agency, body image, and body representation) can be generated. Then, similar psychometric methods can be used with this new data to sort items into distinct, unidimensional constructs. In this way, data-driven definitions for psychosocial constructs can be created from the experiences, tasks, and feelings captured by the item set. It is possible that the field's current definitions of embodiment, self-efficacy, etc. could be modified or expanded through such an approach, with potential implications in broader areas of psychology, phenomenology, and philosophy.

### Limitations

Our study participants were English speakers residing in the United States. Thus, the generalizability of our study is limited to English language speakers. The PEM should be translated into other languages, the translated versions should be cognitively tested, and the findings should be replicated with translated versions. Our sample consisted of predominantly Veteran males and persons with unilateral amputation. Our sample of women and those with bilateral amputation was small and may not have had the power to detect a small magnitude of DIF. In addition, the women included in the study were predominantly non-Veterans, so differences between sexes may have been influenced by Veteran status. While we provided scoring tables in supplemental materials, this scoring method can only be used to calculate summary scores for when there is no missing data.

### Future directions

The structural validity and reliability of the PEM scales supports use of these measures in studies assessing the benefits and limitations of various prosthetic technologies. The PEM can be

leveraged to study the impact of newer interventions available in the field of upper limb prosthetic research, such as osseointegration, targeted muscle reinnervation, regenerative peripheral nerve interfaces, and peripheral nerve stimulation. For example, future research can compare how sensory restoration techniques affect the sense of embodiment relative to conventional prosthetic devices or how osseointegration affects the intuitiveness of prosthesis use compared to conventional prosthetic sockets. Future studies can also make comparisons to the normative values that can be estimated from the data collected here, improving the interpretability of study results. Future studies can also examine differential item functioning by prosthesis type, features or other factors (e.g. race, ethnicity, education), to identify whether there are differences in item difficulty for different populations. There may be differentiated abilities and needs of these demographic subgroups that have not been explored in our work to date.

## Conclusions

This paper reports on the refinement and psychometric evaluation of the Patient Experience Measure (PEM). In summary, the PEM is a validated tool for assessing psychosocial experiences of upper limb prosthesis users and can be used to compare the experiences of users of various types of prostheses and advanced technologies. The six unidimensional PEM scales are: social interaction, self-efficacy, embodiment, intuitiveness, wellbeing, and self-consciousness. The social interaction, self-efficacy, and embodiment scales had strong person reliability, internal consistency, and test-retest reliability. These scales are recommended in future prosthetics research to enable researchers and clinicians to understand the benefits and limitations of new technologies and interventions. The other three scales require further refinement.

## Supporting information

**S1 Appendix. Detailed description of item revision process.**
(DOCX)

**S2 Appendix. Revised patient experience measure.** Measure used in field testing.
(DOCX)

**S3 Appendix. Final revised patient experience measure.**
(DOCX)

**S4 Appendix. Scoring tables.**
(DOCX)

## Author Contributions

**Conceptualization:** Linda J. Resnik, Melissa A. Clark, Pengsheng Ni.

**Formal analysis:** Linda J. Resnik, Mathew L. Borgia, Melissa A. Clark, Emily Graczyk, Jacob Segil, Pengsheng Ni.

**Funding acquisition:** Linda J. Resnik.

**Investigation:** Linda J. Resnik.

**Methodology:** Linda J. Resnik, Mathew L. Borgia, Melissa A. Clark, Pengsheng Ni.

**Project administration:** Linda J. Resnik.

**Software:** Pengsheng Ni.

**Supervision:** Linda J. Resnik.

**Validation:** Pengsheng Ni.

**Visualization:** Pengsheng Ni.

**Writing – original draft:** Linda J. Resnik, Mathew L. Borgia, Melissa A. Clark, Emily Graczyk, Jacob Segil, Pengsheng Ni.

**Writing – review & editing:** Linda J. Resnik, Mathew L. Borgia, Melissa A. Clark, Emily Graczyk, Jacob Segil, Pengsheng Ni.

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
