## [Decision Letter · Decision Letter 0]

22 Sep 2021

PONE-D-21-25532Structural validity and reliability of the patient experience measure: A new approach to assessing psychosocial experience of upper limb prosthesis usersPLOS ONE

Dear Dr. Resnik,

Thank you for submitting your manuscript to PLOS ONE. After careful consideration, we feel that it has merit but does not fully meet PLOS ONE’s publication criteria as it currently stands. Therefore, we invite you to submit a revised version of the manuscript that addresses the points raised during the review process.

We look forward to receiving your revised manuscript.

Kind regards,

Yih-Kuen Jan, PhD, University of Illinois at Urbana-Champaign

2. In the ethics statement in the Methods and online submission information, please clarify whether consent was written or verbal.  If verbal, please also specify: 1) whether the ethics committee approved the verbal consent procedure, 2) why written consent could not be obtained, and 3) how verbal consent was recorded. If your study included minors, state whether you obtained consent from parents or guardians. If the need for consent or parental consent was waived by the ethics committee, please include this information.

“Department of Veterans Affairs Rehabilitation Research and Development Service

A2936-R and A9264-S”

Reviewers' comments:

Reviewer's Responses to Questions

**Comments to the Author**

1. Is the manuscript technically sound, and do the data support the conclusions?

Reviewer #1: Yes

Reviewer #2: Yes

2. Has the statistical analysis been performed appropriately and rigorously? 

Reviewer #1: Yes

Reviewer #2: Yes

3. Have the authors made all data underlying the findings in their manuscript fully available?

Reviewer #1: Yes

Reviewer #2: Yes

4. Is the manuscript presented in an intelligible fashion and written in standard English?

Reviewer #1: Yes

Reviewer #2: Yes

5. Review Comments to the Author

Reviewer #1: Many experiments have been carried out in this study, which makes the whole research more credible. It is also necessary for the author to improve the prosthetic measurement system

1 In terms of limitations, I hope the research can add the division of age. The thinking modes of teenagers, middle-aged and elderly may be different

2 The measurement of PEM is physically straightforward, but there are a thousand Hamlets in a thousand people's eyes on the psychological level. It will be challenging to determine the final index

3. The suggestion is to add an experiment on how many hours a day to wear prosthetics.

Reviewer #2: This study examined the psychosocial outcomes of a tool for people with upper limb amputation and users of sensory prostheses.

The most important thing that this study adds to the body of science is that the psychosocial effect of this tool was unknown in previous studies.

The research question is well articulated and the writing is fluent and simple. But there are some major and minor revision as below:

One of the biggest problems of this manuscript is the lack of literature review in introduction and discussion.

The next problem is that it is not entirely clear the previous measurement tool examine what aspects?

But data is sufficient, tables are clear

Introduction:

Although main question is addressed in the first paragraph but there is no reference and it is not supported by previous scientific document, then much of the introduction is based on an abstract from a symposium (ref 1)

“Resnik L, Gracyzk, E, Tyler, D., editor. Measuring User Experience of a Sensory Enabled Limb Prosthesis. MEC 2017; Frederickton, New Brunswick, CAN.”

The introduction section is not well written. Not only does the reader not achieve similar work in the field of psychosocial evaluation in any other population, but the reader also does not understand the different aspects of this measure.

I strongly suggest that the introduction be rewritten.

Bold the importance of psychosocial aspects in upper extremity amputees and refer to the previous use of this tool.

The manuscript is too long and you can reduce the length by summarizing introduction.

Method:

Why Participants’ oral consent was obtained and why no written consent was obtained

There is no information (sampling and sample size) about 20 and 18 participants in phase prior to cognitive testing and pilot test.

You have written “Participants who were not prosthesis users …..” While you have mentioned only one person was non-user.

Result:

Result section has been written well but in abstract there is no quantitative description for result. Hence, it is better to change result section in abstract.

Since the length of the manuscript is too long, it seems that some tables can be moved to appendix.

Discussion:

In the discussion, the relationship between the study and past studies is still not well expressed

In the discussion, it can be pointed out that there is limited or no tool to evaluate the psychosocial aspects in people with upper limb amputations and people using upper limb prostheses, and tool developing what positive aspects it can have.

Overall summary of scales

I suggest deleting titles “Importance of the work” and “Overall summary of scales”.

Please remove “For example, future research can compare how sensory restoration techniques affect the sense of embodiment relative to conventional prosthetic devices or how osseointegration affects the intuitiveness of prosthesis use compared to conventional prosthetic sockets”

6. PLOS authors have the option to publish the peer review history of their article (what does this mean?). If published, this will include your full peer review and any attached files.

Reviewer #1: **Yes: **Chi-Wen Lung

Reviewer #2: **Yes: **Monireh Ahmadi Bani

---

## [Author Response · Author response to Decision Letter 0]

5 Oct 2021

Please see the detailed response to review document attached. This document has formatting that may be easier to follow. Text from that document is copied below.

Thank-you for the thoughtful review comments. We have responded to each comment below.

Reviewer #1: Many experiments have been carried out in this study, which makes the whole research more credible. It is also necessary for the author to improve the prosthetic measurement system

1 In terms of limitations, I hope the research can add the division of age. The thinking modes of teenagers, middle-aged and elderly may be different

Thank you for this comment. We agree that there may be differential item function by younger or middle age participants. We had only 2 participants under 20 years old in our study. Given the distribution of age in our sample, we decided that the most robust comparisons of differential item functioning would be for persons 65 and over as compared to younger. Future studies can compare PEM scores by additional divisions of age group. 

2 The measurement of PEM is physically straightforward, but there are a thousand Hamlets in a thousand people's eyes on the psychological level. It will be challenging to determine the final index

We appreciate this comment. Certainly, the measurement of psychological experiences is a challenge across many fields of research including rehabilitation. We acknowledge that other measures exist that measure some similar constructs. However, no other measures assess social interactions with the prosthesis or intuitiveness. Furthermore, no existing measure addresses all of the constructs in the PEM.

3. The suggestion is to add an experiment on how many hours a day to wear prosthetics.

The reviewer raises an interesting suggestion, but one which we believe is outside the scope of the current paper. We are planning a second paper that utilizes the PEM to make several comparisons- including differences by hours of prosthesis wear. We also believe that it will be interesting to examine PEM scores by years of prosthesis use and by type of prosthesis use. 

Reviewer #2: This study examined the psychosocial outcomes of a tool for people with upper limb amputation and users of sensory prostheses. The most important thing that this study adds to the body of science is that the psychosocial effect of this tool was unknown in previous studies. The research question is well articulated and the writing is fluent and simple. But there are some major and minor revision as below:

One of the biggest problems of this manuscript is the lack of literature review in introduction and discussion.

We appreciate the reviewers desire to see more literature review as well as a shorter introduction. We have removed some text related to the development of the initial measure and revised the introduction with more references to the literature. 

We added a few sentences to the discussion, but do not feel that more is needed. Respectfully, we believe that our discussion section includes a robust discussion of the content of the PEM scales, the hierarchy of item difficulty, and how the items align with prior theories. Given the novelty of our work and the lack of existing metrics, we think that this is the most pertinent literature to discuss. However, we would be open to Reviewer suggestions for specific elements missing from the discussion.

The next problem is that it is not entirely clear that the previous measurement tool examine what aspects?

Line 89 (clean version) states, “The original scales addressed the key constructs of self-efficacy, embodiment, body image, efficiency of the prosthesis, and social touch.” Furthermore, Table 3 presents the original items used in the previous measurement tool, how the items were altered for field testing, and the final PEM Subscale to which each item was assigned. 

But data is sufficient, tables are clear

Thank you

Introduction:

Although main question is addressed in the first paragraph but there is no reference and it is not supported by previous scientific document, then much of the introduction is based on an abstract from a symposium (ref 1) “Resnik L, Gracyzk, E, Tyler, D., editor. Measuring User Experience of a Sensory Enabled Limb Prosthesis. MEC 2017; Frederickton, New Brunswick, CAN.”

Thank you, we substituted citations here to two peer-reviewed manuscripts that reported on the use of the measure.

The introduction section is not well written. Not only does the reader not achieve similar work in the field of psychosocial evaluation in any other population, but the reader also does not understand the different aspects of this measure. I strongly suggest that the introduction be rewritten. Bold the importance of psychosocial aspects in upper extremity amputees and refer to the previous use of this tool. The manuscript is too long and you can reduce the length by summarizing introduction.

Thank you for these suggestions. We have reworked the introduction, focusing on the importance of psychosocial constructs in upper limb amputation and clarifying the aspects measured by the original version of the PEM.

Methods:

Why Participants’ oral consent was obtained and why no written consent was obtained.

We have revised the text, which now reads, “All phases of the study were approved by the appropriate institutional review boards. All participants received mailed information about the study and gave verbal informed consent.” We added that, “The Institutional Review Board approved use of verbal consent because the study was a nationwide telephone survey; verbal consent (or lack of consent) was documented by the telephone interviewer at the time of telephone contact with the participant.” 

There is no information (sampling and sample size) about 20 and 18 participants in phase prior to cognitive testing and pilot test.

Sample sizes of cognitive testing (N=20) and pilot testing (N=18) are clearly defined and characteristics of the samples are shown in Table 1. Given the Reviewer comment, we added text in the methods section regarding the sufficiency of sample sizes: “There are no definitive rules for determining the appropriate sample size for cognitive interviewing and pilot testing. We conducted cognitive-based interviews until we were unable to identify any new content related to the PEM target constructs, or any major problems with particular questions, response options, or instrument instructions. The sample size for the pilot test was based on available resources and study timeline.

You have written “Participants who were not prosthesis users …..” While you have mentioned only one person was non-user.

Thank you, we have corrected the tenses in the sentence to reflect that – in the cognitive sample – there was only one nonuser.

Result:

Result section has been written well but in abstract there is no quantitative description for result. Hence, it is better to change result section in abstract.

We added brief quantitative results to the abstract as follows, ”After removal of 2 items during Rasch analyses, the overall model fit was acceptable (CFI=0.973, TLI=0.979, RMSEA=0.038).The social interaction, self-efficacy and embodiment scales had strong person reliability (0.81, 0.80 and 0.77, respectively), Cronbach’s alpha (0.90, 0.80 and 0.71), and intraclass correlation coefficients (0.82, 0.85 and 0.74).” 

Since the length of the manuscript is too long, it seems that some tables can be moved to appendix.

We appreciate the Reviewer’s suggestion. One of the reasons we have submitted this paper to PLOS One is that there are no stringent word requirements or limitations on tables and figures. We believe that our current tables are useful to the reader and would prefer to retain them. However, if the Reviewer and Editor agree, we can move Table 1 (characteristics of the cognitive and pilot sample) to the Appendix.

Discussion:

In the discussion, the relationship between the study and past studies is still not well expressed

Thank you for this comment. We have added a sentence to explain the relationship between this study and the prior studies that utilized the PEM but did not examine its psychometric properties. We added the following sentence, “While prior work utilized an earlier prototype of the PEM measure to study the psychosocial experience of persons with sensory enabled prostheses, this is the first study that examined the psychometric properties of the PEM measure.”

In the discussion, it can be pointed out that there is limited or no tool to evaluate the psychosocial aspects in people with upper limb amputations and people using upper limb prostheses, and tool developing what positive aspects it can have.

We agree that this is one of the most important contributions of our work as pointed out in paragraph 1 of the discussion section. To emphasize this point we added the following sentence, “Thus, our work addresses an important gap in measuring psychosocial experiences in persons using upper limb prostheses.”

Overall summary of scales

I suggest deleting titles “Importance of the work” and “Overall summary of scales”.

We have deleted these titles as requested. 

Please remove “For example, future research can compare how sensory restoration techniques affect the sense of embodiment relative to conventional prosthetic devices or how osseointegration affects the intuitiveness of prosthesis use compared to conventional prosthetic sockets”

We believe that this is an important area for future research and a good concrete example of how the PEM can be used. Therefore, we would like to retain this example.

---

## [Decision Letter · Decision Letter 1]

10 Nov 2021

PONE-D-21-25532R1Structural validity and reliability of the patient experience measure: A new approach to assessing psychosocial experience of upper limb prosthesis usersPLOS ONE

Dear Dr. Resnik,

Thank you for submitting your manuscript to PLOS ONE. After careful consideration, we feel that it has merit but does not fully meet PLOS ONE’s publication criteria as it currently stands. Therefore, we invite you to submit a revised version of the manuscript that addresses the points raised during the review process.

We look forward to receiving your revised manuscript.

Kind regards,

Yih-Kuen Jan, PhD

Academic Editor

PLOS ONE

Journal Requirements:

Reviewers' comments:

Reviewer's Responses to Questions

**Comments to the Author**

1. If the authors have adequately addressed your comments raised in a previous round of review and you feel that this manuscript is now acceptable for publication, you may indicate that here to bypass the “Comments to the Author” section, enter your conflict of interest statement in the “Confidential to Editor” section, and submit your "Accept" recommendation.

Reviewer #1: All comments have been addressed

Reviewer #2: All comments have been addressed

2. Is the manuscript technically sound, and do the data support the conclusions?

Reviewer #1: Yes

Reviewer #2: Yes

3. Has the statistical analysis been performed appropriately and rigorously? 

Reviewer #1: Yes

Reviewer #2: Yes

4. Have the authors made all data underlying the findings in their manuscript fully available?

Reviewer #1: Yes

Reviewer #2: Yes

5. Is the manuscript presented in an intelligible fashion and written in standard English?

Reviewer #1: Yes

Reviewer #2: Yes

6. Review Comments to the Author

Reviewer #1: Those comments have been addressed where appropriate, either in the analysis above or in an individual reply to the exporter concerned.

Reviewer #2: Thank you for revise the previous comments well. The manuscript looks good, though it still needs some minor revision.

abstract well written

Introduction corrections are acceptable

In the method section, small changes are needed, which are as follows:

What do you mean by table 1 where the type of prosthesis is mentioned?

If possible, state the length of time since the amputation. The duration of prosthesis use can also be beneficial.

If possible, summarize the factor analysis and rasch analysis. Written too long.

The number of titles in the method section is high. You can, for example, use a title for the two sections Floor and ceiling and Transformation and scoring.

In Table 2, when the Military option is zero for everyone, it is better to delete this row altogether.

I do not understand anything from Table 2 in general? For example, for Body-powered, the number 155 is written with a standard deviation of 73, I do not understand exactly what it means. please clarify.

thanks

7. PLOS authors have the option to publish the peer review history of their article (what does this mean?). If published, this will include your full peer review and any attached files.

Reviewer #1: **Yes: **Chi-Wen Lung

Reviewer #2: No

---

## [Author Response · Author response to Decision Letter 1]

17 Nov 2021

Journal Requirements:

We have reviewed the reference list and are not aware of any references that have been retracted.

Review Comments to the Author

Reviewer #1: Those comments have been addressed where appropriate, either in the analysis above or in an individual reply to the exporter concerned.

Thank you – we are glad the revisions have addressed all concerns.

Reviewer #2: Thank you for revising the previous comments well. The manuscript looks good, though it still needs some minor revision. abstract well written, Introduction corrections are acceptable.

Thank you, we appreciate the Reviewer’s feedback.

In the method section, small changes are needed, which are as follows:

What do you mean by table 1 where the type of prosthesis is mentioned?

Table 1 reports the ’primary prosthesis type’ that is used for the pilot and cognitive samples. We elected to name the category this way because some participants have multiple types of prostheses. The primary type is the type that is used most often. We believe the different types listed (Body-powered, myoelectric, hybrid, etc) will be familiar categories and that this is useful descriptive information about the sample of persons with amputation included in this study. We added a definition of “primary” to the table legend.

If possible, state the length of time since the amputation. The duration of prosthesis use can also be beneficial.

Table 2 provides time since amputation. We did not collect this data for participants in the pilot and cognitive sample. We do not have data on the number of years of prosthesis use.

If possible, summarize the factor analysis and rasch analysis. Written too long.

We agree that the results of the factor and Rasch analyses are long. We deleted some text and referred to information available in tables, but are reluctant to remove more key information that explains our approach. 

The number of titles in the method section is high. You can, for example, use a title for the two sections Floor and ceiling and Transformation and scoring.

Thank-you for this suggestion. We have combined subsection titles as recommended. We also removed EFA results and CFA results and left these under the overall section ‘Factor Analysis’. 

In Table 2, when the Military option is zero for everyone, it is better to delete this row altogether.

We have deleted this row at the reviewer’s request. 

I do not understand anything from Table 2 in general? For example, for Body-powered, the number 155 is written with a standard deviation of 73, I do not understand exactly what it means. please clarify.

Table 2 presents continuous variables as mn(sd) and categorical variables as N(%). These headings are provided within the table. For the example provided, the number of body-powered users (a categorical variable) was 155, which was 73.5% of the sample.

---

## [Decision Letter · Decision Letter 2]

13 Dec 2021

Structural validity and reliability of the patient experience measure: A new approach to assessing psychosocial experience of upper limb prosthesis users

PONE-D-21-25532R2

Dear Dr. Resnik,

We’re pleased to inform you that your manuscript has been judged scientifically suitable for publication and will be formally accepted for publication once it meets all outstanding technical requirements.

Kind regards,

Yih-Kuen Jan, PhD

Academic Editor

PLOS ONE

Additional Editor Comments (optional):

Reviewers' comments:

Reviewer's Responses to Questions

**Comments to the Author**

1. If the authors have adequately addressed your comments raised in a previous round of review and you feel that this manuscript is now acceptable for publication, you may indicate that here to bypass the “Comments to the Author” section, enter your conflict of interest statement in the “Confidential to Editor” section, and submit your "Accept" recommendation.

Reviewer #1: All comments have been addressed

Reviewer #2: All comments have been addressed

2. Is the manuscript technically sound, and do the data support the conclusions?

Reviewer #1: Yes

Reviewer #2: Yes

3. Has the statistical analysis been performed appropriately and rigorously? 

Reviewer #1: Yes

Reviewer #2: Yes

4. Have the authors made all data underlying the findings in their manuscript fully available?

Reviewer #1: Yes

Reviewer #2: Yes

5. Is the manuscript presented in an intelligible fashion and written in standard English?

Reviewer #1: Yes

Reviewer #2: Yes

6. Review Comments to the Author

Reviewer #1: Those comments have been addressed where appropriate, either in the analysis above or in an individual reply to the exporter concerned.

Reviewer #2: thanks for choosing me as a reviewer and thanks the authors for their careful edition. I think the manuscript is ready to publish.

7. PLOS authors have the option to publish the peer review history of their article (what does this mean?). If published, this will include your full peer review and any attached files.

Reviewer #1: **Yes: **Chi-Wen Lung

Reviewer #2: **Yes: **Dr. Monireh Ahmadi Bani

---

## [Editor Report · Acceptance letter]

16 Dec 2021

PONE-D-21-25532R2 

Structural validity and reliability of the patient experience measure: A new approach to assessing psychosocial experience of upper limb prosthesis users 

Dear Dr. Resnik:

I'm pleased to inform you that your manuscript has been deemed suitable for publication in PLOS ONE. Congratulations! Your manuscript is now with our production department. 

Kind regards, 

on behalf of

Dr. Yih-Kuen Jan 

Academic Editor

PLOS ONE